# GUN1 and Plastid RNA Metabolism: Learning from Genetics

**DOI:** 10.3390/cells9102307

**Published:** 2020-10-16

**Authors:** Luca Tadini, Nicolaj Jeran, Paolo Pesaresi

**Affiliations:** Department of Biosciences, University of Milan, 20133 Milan, Italy; luca.tadini@unimi.it (L.T.); nicolaj.jeran@unimi.it (N.J.)

**Keywords:** GUN1, RNA polymerase, transcript accumulation, transcript editing, retrograde signaling

## Abstract

GUN1 (genomes uncoupled 1), a chloroplast-localized pentatricopeptide repeat (PPR) protein with a C-terminal small mutS-related (SMR) domain, plays a central role in the retrograde communication of chloroplasts with the nucleus. This flow of information is required for the coordinated expression of plastid and nuclear genes, and it is essential for the correct development and functioning of chloroplasts. Multiple genetic and biochemical findings indicate that GUN1 is important for protein homeostasis in the chloroplast; however, a clear and unified view of GUN1′s role in the chloroplast is still missing. Recently, GUN1 has been reported to modulate the activity of the nucleus-encoded plastid RNA polymerase (NEP) and modulate editing of plastid RNAs upon activation of retrograde communication, revealing a major role of GUN1 in plastid RNA metabolism. In this opinion article, we discuss the recently identified links between plastid RNA metabolism and retrograde signaling by providing a new and extended concept of GUN1 activity, which integrates the multitude of functional genetic interactions reported over the last decade with its primary role in plastid transcription and transcript editing.

## 1. Introduction

The GUN1 (genomes uncoupled 1) protein is a pentatricopeptide repeat (PPR)-containing protein that localizes to plastids and relays signals to the nucleus after exposure to either norflurazon (NF) or lincomycin (Lin) treatment [1,2]. Although these inhibitors block distinctly different processes—NF inhibits carotenoid biosynthesis by non-competitively binding to phytoene desaturase [3], while Lin is a plastid-specific inhibitor of 70S ribosomes and plastid translation [4]—both drug treatments have been shown to reduce levels of plastid transcripts, indicating that a GUN1-dependent pathway is triggered by perturbation of transcription in the plastids [5]. The involvement of plastid transcription in signaling to the nucleus was initially deduced from the observation that treatment of barley seedlings with tagetitoxin, an inhibitor of the plastid-encoded RNA polymerase (PEP; [6]), decreased transcription of members of the nuclear *RbcS* and *Lhcb1* gene families, without altering plastid DNA replication [7]. Later studies have shown that loss of sigma factor 2 (SIG2) or 6 (SIG6)—each of which is utilized by the PEP to transcribe specific sets of plastid genes [8,9,10]—also activates retrograde signaling [11]. The fact that nuclear gene expression is restored in *sig2 gun1* and *sig6 gun1* double mutants has provided further support for the key role of plastid transcription in triggering a GUN1-dependent retrograde response [11].

Here, we propose a new model for GUN1 function in plastids and GUN1-mediated retrograde communication, which integrates the interactions observed between GUN1 and the machinery that controls plastid protein homeostasis with the recently discovered role of GUN1 in plastid RNA metabolism [12,13,14,15,16]. The many additive phenotypic effects and the very few suppressor phenotypes observed in higher-order (compound) mutants containing *gun1* together with mutations affecting various aspects of plastid protein homeostasis, sugar sensing, and plastid osmosis can all be interpreted as pleiotropic phenotypic effects of a primary alteration in plastid transcription and plastid transcript editing attributable to the lack of the GUN1 protein under conditions that would otherwise trigger plastid-to-nucleus retrograde communication.

## 2. GUN1 and the “*Δ-rpo* Phenotype”

GUN1 does not appear to bind directly to RNAs [17], unlike typical PPR proteins [17]. But it is targeted to an organelle, the chloroplast, and has recently been reported by two independent laboratories to influence plastid transcript accumulation [13,14,15]. In particular, following perturbation of PEP activity, a peculiar pattern of plastid transcription, designated as the “*Δ-rpo* phenotype”, is typically observed in different plant species [14,18,19,20]. For example, Arabidopsis seedlings lacking proteins required for PEP-mediated transcription and regulation, such as sigma factors (*sig6*; [21]), PEP-associated proteins (*pap2* and *pap8*; [22]), and the plastid redox insensitive 2 (*prin2*) protein [23,24], show a decrease in the expression of genes, coding for subunits of the photosynthetic apparatus (*petB*, *psaB*, *psbA*, *psbB*, *psbC*, *psbD*, *rbcL*), which are normally transcribed by PEP (see Figure 1). Conversely, the expression levels of nucleus-encoded RNA polymerase (NEP)-dependent transcripts of genes, encoding the core subunits of the PEP enzymes (*rpoA*, *rpoB*, *rpoC1*), ribosomal subunits (*rps2* and *rps15*), y*cf1* (translocon at inner envelope membrane of chloroplasts 214), and *clpP1* (a subunit of the 350-kDa chloroplast Clp complex), increase significantly (Figure 1). A similar plastid gene expression pattern is observed in seedlings impaired in plastid transcript maturation (*pdm1*; [25]) and in seedlings grown in the presence of Lin (Col-0 + Lin) or altered in plastid protein synthesis by depletion of plastid ribosomal proteins [14]. Strikingly, *gun1* cotyledons have shown a limited increase in the accumulation of NEP-dependent transcripts when grown in the presence of Lin (*gun1* + Lin, [14]; see also Figure 1) and when the *gun1* mutation is introgressed into genetic backgrounds lacking plastid ribosomal subunits [14]. Furthermore, the same set of NEP-dependent transcripts is highly downregulated after treatment of *gun1* seedlings with NF (*gun1* + NF), but their levels do not exhibit major changes in Col-0 + NF and *gun5* + NF seedlings, implying a specific role for GUN1 in NEP-dependent transcript accumulation (Figure 1). Recently, a similar observation has been reported based on a comparison of the plastid gene expression profiles of *cue8* (chlorophyll a/b-binding protein-underexpressed 8) and *cue8 gun1* mutants: i.e., the “*Δ-rpo* phenotype” characteristic of *cue8* seedlings is almost completely abolished in *cue8 gun1* cotyledons [13].

The “*Δ-rpo* phenotype” is thought to be part of the regulatory mechanisms that serve to modulate the activities of NEP and PEP enzymes during plastid differentiation and in the course of responses of mature chloroplasts to environmental cues [27,29,30,31]. The physical interaction of GUN1 with NEP, as revealed by co-immunoprecipitation studies and bimolecular fluorescence complementation (BiFC) assays [14], seems to lie at the basis of the GUN1-mediated accumulation of NEP-dependent plastid transcripts following activation of retrograde communication, possibly favored by decreased competition for template binding and/or increased availability of dNTPs. As an alternative explanation, Loudya et al. proposed that GUN1 sustained chloroplast DNA (cpDNA) replication under specific conditions, as demonstrated by copy numbers of cpDNA, which were reduced by half in *cue8 gun1* chloroplasts with respect to the single mutants and wild-type plants [13].

## 3. GUN1 and Plastid RNA Editing

Like several other PPR proteins (for a review, see [32]), GUN1 has also been reported to influence plastid RNA editing [12], i.e., the enzymatic conversion of specific cytidines (Cs) to uridines (Us) in mRNAs, such that the information in mature RNAs deviates from that encoded in the plastid genome. The link between plastid retrograde signaling and plastid RNA editing was initially reported by Kakizaki et al. [33]. The authors demonstrated that RNA editing in plastids was indeed affected under conditions, such as Lin and NF treatments, that triggered plastid-to-nucleus signaling. In particular, reduced RNA editing levels were observed in *rps14* transcripts and RNAs encoding NAD(P) dehydrogenase (NDH) subunits when Arabidopsis seedlings were treated with either of these drugs. However, direct evidence for a causal relationship between altered RNA editing and plastid signaling was lacking, and the proteins involved remained unknown. This issue has now been clarified by the recent demonstration that GUN1 interacts with an essential component of the plant RNA editosome—multiple organellar RNA editing factor 2 (MORF2)—and affects the editing of multiple plastid RNA sites during retrograde signaling. In particular, *gun1* seedlings treated with NF affect the RNA editing levels at 11 sites in plastids, indicating the high specificity of GUN1 for target RNAs. Intriguingly, all the edited sites are in NEP-dependent transcripts encoding subunits of the ATP-dependent Clp protease, the NDH complex, the ribosomes, photosystem II, and core subunits of PEP. Furthermore, treatment of *gun1* mutants with either NF or Lin leads to similarly increased RNA editing levels only at the *rps12-i-58* site, while comparable decreases are observed for *rpoC1-488*, *psbZ-50,* and *rpoB-551* editing sites [12]. RNA editing is usually associated with amino-acid substitutions in protein-coding sequences or is used to create start/stop codons, thus serving as a correction mechanism for otherwise defective transcripts. As a matter of fact, mutants with altered RNA editing in plant organelles exhibit defects in development and growth, including pale and albino phenotypes [34,35,36,37]. In this specific case, GUN1 is responsible for the exchange of three highly conserved amino-acid residues in the β subunit of the PEP core, known to be the catalytic site of the enzyme, and one in the *rpoC1*-encoded β’ subunit, also part of the PEP core but with unknown function. Therefore, it is tempting to hypothesize that changes in RNA editing levels for *rpoB* and *rpoC1* ultimately affect the activity of PEP. This, together with the impairment of GUN1-dependent regulation of NEP activity, might be the primary cause of the altered plastid transcript accumulation patterns observed in *gun1* cotyledons upon NF- or Lin-based stimulation of plastid communication with the nucleus ([14]; see also Figure 1).

## 4. Genetic Evidence for GUN1′s Interactions with the Plastid Protein Homeostasis Machinery

Genetic evidence indicates that GUN1 plays an essential role during the early stages of chloroplast development in response to functional impairment of plastid gene expression, plastid protein import, plastid protein degradation, sugar sensing, or maintenance of plastid osmosis (see Table 1 and Table 2 for details). This notion is based on the additive phenotypic effects observed when the *gun1* mutation is introgressed by manual crosses into genetic backgrounds, exhibiting defects in plastid transcription, transcript maturation and editing, plastid protein synthesis, import or degradation, or sugar sensing. Among the 31 *gun1*-containing higher-order mutants listed in Table 1 and Table 2, 20 are characterized by additive phenotypic effects with respect to the single mutants (see also Figure 2). These include viable pale-green plants with reduced growth and photosynthetic performance, albino seedlings unable to grow autotrophically, and one embryo-lethal combination. These enhanced phenotypes, leading in some cases to non-viable double mutants, reveal the importance of GUN1 function in the plastid. They confirm that GUN1 activity indeed supports plant development by optimizing chloroplast biogenesis—and hence photosynthetic efficiency—even when plastid gene expression is impaired by genetic modifications.

In eight genetic backgrounds, the *gun1* mutation does not exacerbate the original mutant phenotype. The lack of any additive effect in *gun1 ftsh1-1* and *gun1 ftsh8-1* cotyledons and leaves can be ascribed to the fact that defective FtsH1 can be complemented by the major FtsH5 (type A) subunit, while FtsH8 can be replaced by the major FtsH2 (type B) subunit of the heteromeric filamentation temperature-sensitive (FtsH) metalloprotease associated with thylakoids (for a review, see [45]). Indeed, unlike *ftsh2* and *ftsh5*, single *ftsh1-1* and *ftsh8-1* mutants do not display any visible phenotype [14], indicating that lack of either product does not perturb the activity of the FtsH metalloprotease. This can explain why GUN1-mediated retrograde signaling is not required in *ftsh1-1* or *ftsh8-1*. Moreover, no additive effects are observed when the *gun1* mutation is combined with *gun2*, *gun4,* and *gun5* single mutants [46,47]. Indeed, the double mutants *gun1 gun2*, *gun1 gun4,* and *gun1 gun5* accumulate chlorophylls to very similar levels and display growth rates comparable to those of *gun2*, *gun4,* and *gun5* single mutants (see Table 2). Conversely, *gun2 gun4* and *gun4 gun5* double mutants are characterized by a more extreme chlorophyll phenotype than that of the corresponding single mutants. The exacerbation of the single mutant phenotype is even more prominent in *gun2 gun5*, in which chlorophyll is undetectable [46,47]. Overall, these functional interactions allow us to exclude a major role for GUN1 in the tetrapyrrole biosynthetic pathway, which provides chlorophylls and other tetrapyrrole end-products, such as heme, siroheme, and phytochromobilin.

On the other hand, *gun1 prin2-1*, *gun1 bpg2-2*, and *gun1 prps21-1* double mutants are also indistinguishable from the visible phenotypes of *plastid redox insensitive 2-1* (*prin2-1*; [23]), *brassinazole insensitive pale green 2-2* (*bpg2-2*; [40]), and *plastid ribosomal protein s21-1* (*prps21-1*; [17]), respectively. In particular, comparison of *prps21-1* with *gun1 prps21-1* mutants seems to support the existence of an ‘impairment threshold’ in plastid gene expression, below which the activity of GUN1 is not required, or at the least, GUN1 activity is so low that its complete loss does not exacerbate the corresponding double mutant phenotypes. As a matter of fact, the *prps21-1* single mutant accumulates more PEP-dependent (*rbcL* and *psbA*) and NEP-dependent (*rpoA* and *rpl12-3′*) transcripts than the *prpl11-1* single mutant [14], possibly explaining the marked difference between the albino-lethal phenotype of *gun1 prpl11-1* seedlings and the pale-green phenotype of *gun1 prps21-1* plants.

Only in three cases, *gun1*-containing higher-order mutants display an attenuated (suppressor) phenotype with respect to single mutants. For instance, the callus tissue formation observed in the shoot apex of the *msl2 msl3* double mutant is suppressed in the *gun1 msl2 msl3* triple mutant, which results in the formation of green and normally shaped true leaves. Clearly, this genetic interaction highlights a major role of GUN1 protein in chloroplast biogenesis during the switch from leaf cell proliferation to expansion and differentiation [48]. The direct involvement of GUN1 in chloroplast biogenesis and, as a consequence, in cotyledon and leaf greening is further supported by the suppression in the *gun1 sg1* double mutant of the delayed-greening phenotype seen in *sg1* plants, which is itself characterized by newly formed albino leaves that gradually turn green and become fully green by 3 weeks after germination ([38]; see also Table 1). Similarly, the *gun1 prps1-1* double mutant shows less severe bleaching of cotyledons and leaves and increased photosynthetic performance and growth with respect to the *prps1-1* single mutant due to GUN1-dependent control of the accumulation of the plastid ribosomal protein S1 (PRPS1; [17]).

## 5. GUN1: A Major Checkpoint for the Control of Developmental Defects during Chloroplast Biogenesis and Mitigation of the Deleterious Effects of Stress

GUN1 is a very low-abundance protein with a very short half-life and has never been detected in analyses of the plastid proteome. However, its stability and amount increase upon activation of retrograde signaling, as a consequence of the reduction/inhibition of Clp protease activity [51]. Nevertheless, it is reasonable to assume that even under conditions that activate the retrograde signaling pathway(s), GUN1 is unlikely to have a direct regulatory effect on the many highly abundant proteins that make up the plastid protein homeostasis machinery. A more realistic view suggests that most of the genetic interactions observed in the last decade and reported in Table 1 and Table 2 can be ascribed to pleiotropic effects.

The recent findings that point to a primary and direct role of GUN1 in plastid RNA metabolism imply that the protein stimulates NEP activity and alters editing levels of a few NEP-dependent transcripts [12,13,14,15,16] and offer a novel perspective on GUN1 activity. In particular, the decreased accumulation of NEP-dependent transcripts, i.e., transcripts encoding mainly rRNA, tRNA, and housekeeping proteins, and the reduced editing levels of some of them observed in *gun1* seedlings upon either NF or Lin treatments, straightforwardly explain the additive effects observed in *gun1*-containing double mutants with defects in (i) plastid transcription, (ii) plastid transcript maturation and editing, and (iii) plastid translation. Moreover, the reduced accumulation of NEP-dependent *ycf1* transcripts [14], which code for the Tic214 subunit of the 1-MDa TIC (translocon at the inner envelope membrane of chloroplasts) complex involved in protein import into the stroma [52], could explain the enhanced/altered phenotypes of the double mutants *gun1 tic40-4*, *gun1 ppi2*, *gun1 cphsc70-1,* and *gun1 clpc1-1*, all of which are defective in plastid protein import. Similarly, the albino-seedling lethal phenotype of *gun1 ftsh2-3* and the severely variegated cotyledons typical of *gun1 ftsh5-3* could be due to the concomitant alteration of the thylakoid-associated heteromeric metalloprotease FtsH and the stromal Clp protease.

With respect to suppressor phenotypes, gene expression analysis has revealed that the introduction of *gun1* into plants carrying the *sg1* genetic background partially rectifies the imbalance in the expression of chloroplast-related genes caused by the *sg1* mutation [38]. For example, the *gun1 sg1* double mutant shows increased expression of *RbcL* and *accD* (both of which are expressed at very low levels in *sg1*) and much reduced expression of *rpoB*, relative to that in *sg1*. Therefore, it seems that *gun1* can partially correct the imbalance in levels of chloroplast-related genes in *sg1*, thereby suppressing the deleterious phenotypes. Similarly, the increased accumulation of PRPS1 protein observed in the leaky *prps1-1* mutant in the absence of the GUN1 protein (*gun1 prps1-1*; [17]) could be related to the lack of upregulation and editing of *ClpP1* transcripts encoding a subunit of the major plastid stromal protease. As a consequence, PRPS1 may be degraded less efficiently [17,51]. The suppression of the *msl2 msl3* double mutant phenotype seen in *gun1 msl2 msl3* seedlings is more difficult to explain [48]. Certainly, GUN1-mediated retrograde communication is only one of the many regulatory pathways and feedback loops that govern dynamic cell identity decision-making at the plant shoot apex. It may be speculated that the absence of this communication still permits proper leaf differentiation, even in the presence of plastid dysfunctions in the shoot apex.

This large collection of genetic and molecular data related to GUN1 function can now be integrated into a model, which is based on the following lines of evidence (Figure 2):GUN1 is present in very low amounts as long as chloroplast biogenesis proceeds normally, i.e., in the absence of stresses/dysfunction of developing plastids. As a matter of fact, the *gun1* mutant is hardly distinguishable from wild-type plants under optimal chloroplast biogenesis conditions (Figure 2a).GUN1 protein levels increase when stresses and/or alterations of plastid functions occur during chloroplast biogenesis. Under these conditions, NEP activity is favored relative to PEP in the developing chloroplasts. This, together with the ensuing retrograde inhibition of photosynthesis-associated nuclear gene (*PhANGs*) expression, results in pale cotyledons and/or leaves in the best-case scenario. Therefore, the prevention of photo-oxidative damage seems to prevail over the optimal organization of the photosynthetic apparatus and its capacity for light absorption (Figure 2b).Lack of GUN1 disables, at least partially, retrograde signaling and its repressive influence on *PhANGs* and causes major alterations in plastid RNA metabolism, including reduced NEP activity and changes in editing levels of RNAs encoding subunits of the ATP-dependent Clp protease, the NDH complex, the ribosomes, photosystem II, and the core of the PEP enzyme (Figure 2c). When the *gun1* mutation is introgressed into genetic backgrounds with defects in either the plastid protein homeostasis machinery (see as examples *sca3-1*, *sg1*, *prpl11-1*, *prps1-1*, *ftsh5-3*, *prin2-1*, *bpg2-2*, *ftsh1-1*; for further details, refer to Table 1), sugar sensing (*sicy-192*; for further details, refer to Table 2), or plastid osmosis (*msl2 msl3*; for further details, refer to Table 2), the corresponding higher-order mutants show a range of phenotypes. (i) In most cases, an exacerbated phenotype is observed, as a consequence of the fact that *gun1*-associated alterations are added to the impairments caused by the original mutant backgrounds as, for instance, in the case of *sca3-1 gun1-102*, *prpl11-1 gun1-102*, *ftsh5-3 gun1-102* (for further details, see Table 1). (ii) In a few cases, no additive phenotype is detected, either because GUN1 activity is not required under that specific plastid perturbation, or is rather limited, as in the case of *prin2-1 gun1-1*, *bpg2-2 gun1-101*, *ftsh1-1 gun1-102* (for further details, see Table 1). (iii) In a small minority of cases, a suppressor phenotype is observed as a consequence of the ability of *gun1*-associated alterations to mitigate the imbalances caused by the initial mutant backgrounds. This is the case of *sg1 gun1-1* and *prps1-1 gun1-102* double mutants and *msl2 msl3 gun1-9* triple mutant.

Overall, GUN1 seems to modulate photosynthetic efficiency in cotyledons and leaves to minimize the consequences of malfunctions in developing chloroplasts, primarily with a view to preventing, or at least reducing, photo-oxidative damage.

**Figure 2 cells-09-02307-f002:**
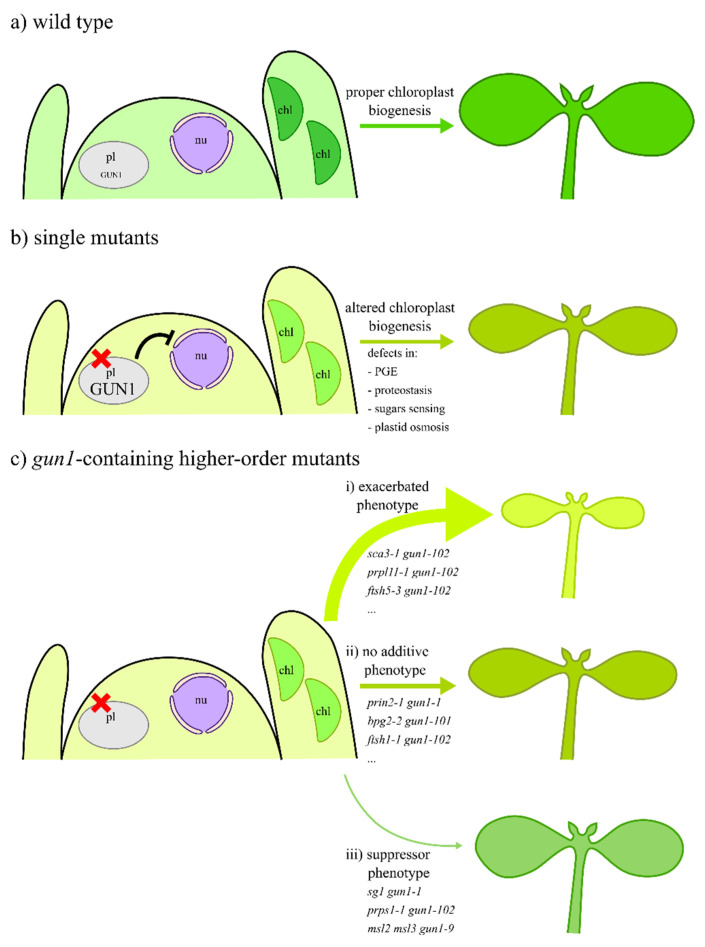
Schematic overview of the role of GUN1 protein during the early stages of chloroplast biogenesis. (**a**) Under optimal conditions, i.e., environmental and genetic conditions, the abundance of the GUN1 protein is rather low, and no GUN1-dependent negative retrograde signal is sent from the developing chloroplasts to the nucleus to downregulate the expression of *PhANGs*. As a result, proper chloroplast biogenesis occurs in cotyledons and leaves. (**b**) Under conditions that alter plastid activity, i.e., genetic defects that impair plastid gene expression (PGE), plastid protein homeostasis (proteostasis), sugar sensing, and plastid osmosis, the abundance of GUN1 protein increases in developing chloroplasts. As a consequence, NEP activity is favored over PEP, changes in plastid RNA editing levels take place, and a GUN1-dependent negative signal reaches the nucleus and reduces expression of *PhANGs*. In this scenario, seedlings show defects in chloroplast development, photosynthetic performance, and growth. (**c**) The importance of the GUN1 protein in chloroplast biogenesis becomes evident when the *gun1* mutation is introgressed into Arabidopsis mutants with defects in PGE, proteostasis, sugar sensing, and maintenance of plastid osmosis. Under these conditions, chloroplast biogenesis is altered, and the GUN1-dependent negative retrograde signal is absent. These conditions, in most of the analyzed cases (20 out of 31, 64.5%), result in exacerbated phenotypes, characterized by a marked reduction in leaf pigmentation and reduced photosynthetic performance and growth. Only in few cases (25.8%), the lack of GUN1 protein fails to cause additive phenotypic effects, while in the remaining three cases (9.7%), lack of GUN1 suppresses the mutant phenotypes. pl, proplastid; nu, nucleus; chl, chloroplast.

## 6. Conclusions and Open Questions

Over the past decade, several labs worldwide have collected important pieces of genetic and molecular information concerning the role of GUN1 during the early stages of chloroplast biogenesis. The very recent data on the involvement of GUN1 in plastid RNA metabolism obtained in our lab, almost concomitantly with the labs of Joanne Chory and Enrique Lopez-Juez, allow us to integrate all the information now available into a unified model, which is schematically depicted in Figure 2 [12,13,14,15,16]. As yet, little can be said about how GUN1 is recruited to modulate transcript accumulation and editing under conditions that trigger retrograde communication. Furthermore, the molecular mechanism(s) that link(s) plastid RNA metabolism to retrograde signaling deserve(s) further study. The changes in editing levels of transcripts encoding subunits of the NDH complex prompt the speculation that alteration of NDH activity could affect the redox state of plastids, thereby triggering retrograde communication. In addition, modification of Clp protease activity, as a consequence of a GUN1-dependent influence on *clpP* RNA metabolism, could be at the basis of the unfolded protein response signaling pathway [39,53,54,55]. Similarly, altered expression of the plastid gene *ycf1* could provide the connection between plastid protein import, cytosolic folding stress, and plastid precursor protein-mediated retrograde communication [14,44]. Further analyses of GUN1 and its involvement in plastid RNA metabolism will provide important clues to the molecular mechanisms at the root of the coordination of plastid-nucleus gene expression. In this context, we suggest the use of GUN1 chimeras expressed under the control of the wild-type GUN1 promoter—which can be obtained by the recently developed CRISPR/Cas9-mediated gene targeting approach [56]—to resolve remaining ambiguities in GUN1 function—introduced, most probably, by the use of *GUN1* over-expressing lines for molecular studies. This strategy, designed in several labs to boost the very low abundance of the GUN1, might have led to the identification of interacting partners that are not essential for its function, as a consequence of the rather sticky nature of the protein, as highlighted by the identification of several protein interactors [57]. In the end, GUN1 may turn out to an “almost” typical plastid PPR protein, as indicated by its involvement in RNA metabolism.

## Figures and Tables

**Figure 1 cells-09-02307-f001:**
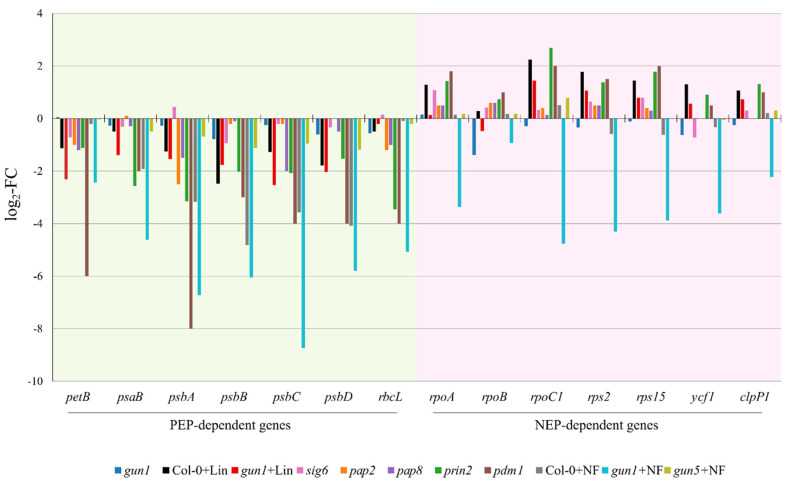
Expression analyses of a subset of PEP- and NEP-dependent plastid genes in different genetic backgrounds and after treatments that alter PEP activity. Values are expressed as the logarithm of the fold-change (log_2_-FC) relative to wild-type seedlings grown under optimal conditions. Data are retrieved from the literature in the case of *sig6* [26], *pap2* and *pap8* [27], *prin2* [23], and *pdm1* (*pigment-deficient mutant 1*) [28] and from the GEO public repository in the case of *gun1*, Col-0 + Lin, *gun1* + Lin (GEO accession GSE5770; [2]), Col-0 + NF, *gun1* + NF, and *gun5* + NF (GEO accession GSE12887;[2]). Lincomycin treatment of wild-type Arabidopsis seedlings (Col-0 + Lin) leads to a drop in PEP-dependent transcript accumulation. As an adaptive response, NEP-dependent transcript levels are increased in plastids, giving rise to what is known as the “*Δ-rpo* phenotype”. Similar behavior is observed in mutants, such as s*ig6*, *pap2*, *pap8*, *prin2,* and *pdm1*, which lack either nucleus-encoded plastid proteins required for PEP activity (SIG6, PAP2, PAP8, PRIN2) or mRNA maturation factors (PDM1). However, *gun1* seedlings grown in the presence of lincomycin (*gun1* + Lin) show an impaired *“Δ-rpo* phenotype”: i.e., NEP-dependent transcripts show a very limited degree of upregulation. Similarly to lincomycin, *gun1* seedlings grown on norflurazon-containing medium (*gun1* + NF) undergo severe repression of PEP-dependent genes, and repression is even more pronounced if NEP-dependent transcripts are considered. On the contrary, Col-0 and *gun5* (*genomes uncoupled 5*) seedlings in the presence of norflurazon (Col-0 + NF and *gun5* + NF) show a wild-type-like plastid gene expression, indicating that the drastically reduced accumulation of PEP- and NEP-dependent transcripts upon NF treatment is a characteristic of the *gun1* genetic background.

**Table 1 cells-09-02307-t001:** Visible phenotypic characteristics of Arabidopsis mutants altered in plastid protein homeostasis and crossed with *gun1*. Arabidopsis mutants affected in plastid transcription, plastid transcript maturation and editing, plastid translation, plastid protein import, and plastid protein degradation have been crossed manually with different *gun1* alleles, including *gun1-1*^a^, *gun1-9*^b^, *gun1-101*^c^, *gun1-102*^d^, *gun1-103*^e^. The phenotypic characteristics of single and higher-order mutants, together with the existence of physical interactions between the corresponding proteins with GUN1, are reported. Superscript a–e letters specify the *gun1* alleles introgressed in the different mutant backgrounds reported in the Table.

Locus	Function	Single Mutant Name and Phenotype	Double Mutant Phenotype	Additive (A) or Suppressor (S) Effect	Physical Interaction	Ref.
**Plastid Transcription**						
At2g24120	NEP: Nucleus- encoded RNA polymerase	*sca3-1*^d^: pale cotyledons; reduced growth	albino-seedling lethal	A	Yes	[14]
At1g08540	SIG2: determines PEP promoter specificity	*sig2-2*^b^*:* pale cotyledons; reduced growth	paler green/yellow cotyledons and young leaves	A	No	[11]
At2g36990	SIG6: determines PEP promoter specificity	*sig6-1*^b^: identical to WT	yellow/white cotyledons	A	No	[11]
At3g18420	SG1: chloroplast-localized, tetratricopeptide repeat-containing protein required for chloroplast development; involved in the regulation of plastid gene expression	*sg1*^a^: slow green1—newly formed albino leaves gradually turn pale-green and are fully green at 3 weeks after germination; reduced growth	the delayed-greening phenotype of the *sg1* single mutant is alleviated; leaves of *sg1 gun1* are of similar green color to the leaves of WT plants	S	No	[38]
At1g10522	PRIN2: regulates PEP activity	*prin2-1*^a^: yellow/white cotyledons; reduced growth	identical to *prin2-1* single mutant	No effect	No	[23]
**Plastid Transcript Maturation/Editing**						
Unknown	Cab-underexpressed 8 (Cue8): chloroplast development	*cue8*^a^: virescent-delayed greening; reduced growth; altered RNA editing	albino-seedling lethal	A	Unknown	[13]
At3g03710	RIF10: exoribonuclease—processing of plastid RNA	*rif10-2*^c^: green cotyledons and pale true leaves; reduced growth	albino-seedling lethal	A	No	[39]
At3g57180	BPG2: regulates ribosomal RNA maturation	*bpg2-2*^c^: pale green/yellow cotyledons	identical to *bpg2-2* single mutant	No effect	No	[40]
At3g06980	RH50: modulates RNA secondary structure	*rh50-1*^d^: identical to WT	marked reduction of growth rate	A	No	[41]
At4g02990	mTERF4: processing of plastid transcripts	*coe1/mterf4*^e^: pale-yellow cotyledons and leaves; reduced growth	more severe pale-yellow phenotype; reduced growth	A	No	[42]
**Plastid Translation**						
At1g17220	FUG1: chloroplast translation initiation factor	*fug1-3*^e^: pale-green cotyledons and leaves; reduced growth	yellow cotyledons and leaves; enhanced reduction of growth rate and photosynthesis performance	A	Yes	[17,43]
At1g32990	PRPL11: plastid ribosomal protein L11	*prpl11-1*^d^: pale-green cotyledons and leaves; reduced growth	albino-seedling lethal	A	No	[17]
At5g30510	PRPS1: plastid ribosomal protein S1	*prps1-1*^d^*:* pale green cotyledons and leaves; reduced growth	less severe pale cotyledons and leaves; increased growth	S	Yes	[17]
At3g27160	PRPS21: plastid ribosomal protein S21	*prps21-1*^d^: pale green cotyledons and leaves; reduced growth	identical to *prps21-1* single mutant	No effect	No	[17]
At5g54600	PRPL24: plastid ribosomal protein L24	*prpl24-1*^d^: pale green cotyledons and leaves; reduced growth	albino-seedling lethal	A	No	[41]
At1g79850	PRPS17: plastid ribosomal protein S17	*prps17-1*^d^: pale green cotyledons and leaves; reduced growth	albino-seedling lethal	A	No	[41]
**Plastid Protein Import**						
At5g16620	Tic40: subunit of the plastid protein import apparatus	*tic40-4*^c^: pale-green cotyledons and leaves; reduced growth	embryo-lethal	A	No	[44]
At4g02510	Toc159: plastid protein import receptor	*ppi2-2*^c^: albino-seedling lethal	embryo-lethal	A	No	[33]
At4g24280	cpHSC70-1: plastid protein import and folding	*cphsc70-1*^c,d^: altered cotyledon and leaf shape; slight variegation	much smaller cotyledons; larger variegation; reduced growth	A	Yes	[14,44]
At5g50920	CLPC1: protein import into chloroplast stroma	*clpc1-1*^c^: pale-green cotyledons and leaves; reduced growth	reduced photosynthetic performance; reduced growth	A	Yes	[44]
**Plastid Protein Degradation**						
At1g50250	FTSH1: subunit of the thylakoid-associated heteromeric FTSH protease	*ftsh1-1*^d^: cotyledons identical to WT	identical to *ftsh1-1* single mutant	No effect	No	[14]
At2g30950	FTSH2: subunit of the thylakoid-associated heteromeric FTSH protease	*ftsh2-3*^b^: pale-green and small cotyledons, reduced growth	albino-seedling lethal	A	No	[14]
At5g42270	FTSH5: subunit of the thylakoid-associated heteromeric FTSH protease	*ftsh5-3*^d^: cotyledons identical to WT	severely variegated cotyledons	A	No	[14]
At1g06430	FTSH8: subunit of the thylakoid-associated heteromeric FTSH protease	*ftsh8-1*^d^: cotyledons identical to WT	identical to *ftsh8-1* single mutant	No effect	No	[14]
At1g49970	CLPR1: subunit of the chloroplastic endopeptidase Clp complex	*clpr1*^c^: pale-green and small cotyledons; reduced growth	albino-seedling lethal	A	No	[39]

RIF10, resistant to inhibition with fosmidomycin 10; mTERF4, mitochondrial transcription termination factor 4; FUG1, fu-gaeri1; Tic40, translocon at the inner envelope membrane of chloroplasts; Toc159, translocon at the outer envelope membrane of chloroplasts; cpHSC70-1, chloroplast heat shock protein 70-1; CLPC1, caseinolytic protease complex component C1; CLPR1, caseinolytic protease complex component R1; FTSH, filamentation temperature sensitive metalloprotease.

**Table 2 cells-09-02307-t002:** Visible phenotypic characteristics of Arabidopsis mutants altered in tetrapyrrole biosynthesis and other functions and crossed with *gun1*. Arabidopsis mutants affected in tetrapyrrole biosynthesis, plastid osmosis, sugar metabolism, and plastid gene expression are crossed manually with different *gun1* alleles, including *gun1-1*^a^, *gun1-9*^b^, *gun1-101*^c^. The phenotypic characteristics of single and higher-order mutants, together with the existence of physical interactions between the corresponding proteins with GUN1, are reported. Superscript a–c letters specify the *gun1* alleles introgressed in the different mutant backgrounds reported in the Table.

Locus	Function	Single Mutant Name and Phenotype	Double Mutant Phenotype	Additive (A), Suppressor (S) Effect	Physical Interaction	Ref.
**Tetrapyrrole Biosynthesis**						
At2g26670	GUN2: heme oxygenase	*gun2*^a^: long hypocotyl; pale green cotyledons; reduced growth	identical to *gun2*	No effect	No	[46]
At3g59400	GUN4: regulates Mg-chelatase	*gun4*^a^: pale green cotyeldons and leves; reduced growth	identical to *gun4*	No effect	No	[47]
At5g13630	GUN5: ChlH subunit of Mg-chelatase	*gun5*^a^: pale green cotyledons and leaves; reduced growth	identical to *gun5*	No effect	No	[47]
**Other Plastid Functions**						
At5g10490; At1g58200	MSL2 and MSL3: two members of the MscS-like family of mechanosensitive ion channels. They are localized in the plastid envelope and are required for normal plastid size and shape	*msl2 msl3*^b^: enlarged and deformed plastids in the shoot apical meristem; develop a mass of callus tissue at the shoot apex	abolished callus formation at the shoot apex; larger, greener, and more normally shaped true leaves	S	No	[48]
At5g22510	INV-E: a chloroplast-targeted alkaline/neutral invertase that is implicated in the development of the photosynthetic apparatus	*sicy-192*^c^: *sugar-inducible cotyledon yellow-192 mutant*: yellow cotyledons upon treatment with sucrose; gain of function mutant of plastid invertase	enhanced cotyledon phenotype due to a further decrease of chlorophyll content	A	No	[49]
At1g31410	ENF2: a chloroplast-targeted protein similar to bacterial polyamine transporters; important for plastid gene expression	*enf2-1*^a^: *enlarged fil expression domain2 mutant*—mature leaves are pale green, more serrated, and narrower than WT; in less than 1% of cases, *enf2-1* forms needle-like leaves; chloroplast development is delayed	albino-seedling lethal	A	No	[50]

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
