# Peer review of "GUN1 and Plastid RNA Metabolism: Learning from Genetics"

_cells, 2020, doi:10.3390/cells9102307_

Round 1

Reviewer 1 Report

In this Opinion Article, the authors review and discuss the connexions between plastid RNA metabolism and plastid-to-nucleus retrograde signalling proposing a new model to integrate the role of GUN1 in this communication pathway.

The paper is well written and organized which is critical, since it is very dense and sometimes it is not easy to follow it if you are not an expert in the subject. Based in a complete an actualised bibliographic revision, the authors analyse and contextualise the latest discoveries in the topic and points out the direction that future research should take which is what you expect in a Review or Opinion Article.

Author Response

We are grateful to Reviewer 1 for the very positive comments made on our Opinion Article. As he/she did not request any changes to the text, there is no need to upload a modified version of our manuscript or a point-by-point response to reviewer’s comments

With my best Regards

Paolo Pesaresi

Reviewer 2 Report

In their manuscript, Tadini et al. discuss the potential implication of a chloroplast-localized pentatricopeptide repeat protein GUN1 in retrograde signalling and put into larger context recent discoveries of its genetic and physical interactions. This is a timely and well-written review and I have only few suggestions, which hopefully further improve the manuscript.

Minor points

Introduction

(page 1, lines 36-39) How retrograde signaling activation provides support to a key role of GUN1 is not evident at all. It would be great to explain a bit more the rationale behind the connection that the authors make.

(1, 39-42) This sentence is very difficult for a non-specialist to make sense of. First, the significance in the context of the arguments is unclear. Second, it is not explained what flu seedling are and how they are important to the overall picture, especially given that they are not mentioned anywhere else in the manuscript. Third, the word “abrogate” is used in law, but not so much in biosciences (wouldn’t "cancel"/"abolish"/"stop" work just as well?).

(10, 222) “nicely explain” — An emotionally neutral word (e.g., straightforwardly) would be more appropriate.

(12, Fig.2c) Adding a few examples (perhaps even to the figure) of the double mutants in question would be helpful.

Author Response

We are grateful to Reviewer 2 for the very positive comments made on our Opinion Article and for the corrections he/she requested. A point-by-point response to reviewer’s comments is copied below and a revised version of our manuscript with track-changes is attached to this message

With my best Regards

Paolo Pesaresi

(page 1, lines 36-39) How retrograde signaling activation provides support to a key role of GUN1 is not evident at all. It would be great to explain a bit more the rationale behind the connection that the authors make.

Reply: We have clarified this aspect by adding the following sentence: “The fact that nuclear gene expression is restored in sig2 gun1 and sig6 gun1 double mutants provided further support for the key role of plastid transcription in triggering a GUN1-dependent retrograde response [11]”.

(1, 39-42) This sentence is very difficult for a non-specialist to make sense of. First, the significance in the context of the arguments is unclear. Second, it is not explained what flu seedling are and how they are important to the overall picture, especially given that they are not mentioned anywhere else in the manuscript. Third, the word “abrogate” is used in law, but not so much in biosciences (wouldn’t "cancel"/"abolish"/"stop" work just as well?).

Reply: We agree with the reviewer that this sentence is, indeed, too difficult for non-specialist readers and that does not add much to the context. Therefore, to make the message easier to understand we prefer to remove the entire sentence

(10, 222) “nicely explain” — An emotionally neutral word (e.g., straightforwardly) would be more appropriate.

Reply: As recommended, we have exchanged “nicely” with “straightforwardly”.

(12, Fig.2c) Adding a few examples (perhaps even to the figure) of the double mutants in question would be helpful.

Reply: We have added few examples of the double mutants in the text at page 11 and in panel C of Figure 2 at page 12, as requested